# Research on the Regulation of Algorithmic Price Discrimination Behaviour of E-Commerce Platform Based on Tripartite Evolutionary Game

**Jianjun Li, Xiaodi Xu *** and **Yu Yang**

School of Computer and Information Engineering, Harbin University of Commerce, Harbin 150028, China
*   Correspondence: xxd@s.hrbcu.edu.cn; Tel.: +86-152-4602-3344

**Abstract:** With the development of the digital economy, the algorithms and big data technologies of e-commerce platforms have gradually turned into double-edged swords. While realising personalised recommendations, they also provide information technology support for the use of algorithmic prices to discriminate and extract residual value from consumers. Consumers frequently use Black Cat and third-party media to complain, resulting in a significant negative impact. Therefore, in order to regulate algorithmic price discrimination, using e-commerce platforms, local governments and consumers act as game subjects, taking an evolutionary game approach. We analyse the impact of different situations and factors on the system's evolutionary stability strategy and conduct its verification via simulation experiments. This study shows that several measures, such as increasing cooperation with the media; establishing clear regulatory rules to reduce the extent of algorithmic price discrimination and the grey revenue of e-commerce platforms; establishing a long-term mechanism for consumer feedback; improving rewards and punishments to increase the probability of successful regulation and penalties by local governments; sharing information to reduce the cost of consumer regulation; and setting reasonable bonus thresholds based on government revenue and consumer regulation costs, can effectively regulate algorithmic price discrimination and promote the sustainable development of e-commerce platforms.

**Keywords:** e-commerce platforms; algorithmic price discrimination; behavioural regulation; evolutionary games



## 1. Introduction

In the context of the digital economy, e-commerce platforms are able to analyse consumer behaviour data to understand their interest preferences to achieve personalised recommendations, thereby increasing consumer stickiness and order turnover [1]. However, they also provide data to support the use of algorithms for price discrimination [2]. According to the results of a survey conducted by the Beijing Consumers' Association in 2022, more than 86.91% of consumers have been "discriminated" against by e-commerce platforms based on algorithmic pricing. The platforms discriminate against consumers based on their frequency of consumption and personal information in order to extract residual value from consumers via algorithmic pricing [3].

Algorithmic price discrimination has gradually attracted national attention. The Anti-Monopoly Committee of the State Council in its Anti-Monopoly Guidelines for the Platform Economy focuses on the certification factors for algorithmic price discrimination; the State Administration for Market Regulation of China incorporated algorithmic price discrimination into the Provisions on Administrative Punishment for Price Violations in January 2021, which impose fines and revoke business licenses for e-commerce platforms that violate the provisions. Although government regulators have regulated price discrimination on platforms, platforms still have a fluke mentality and repeatedly use algorithms to discriminate on price in order to make high profits. A search for price discrimination on the

Black Cat complaint platform, shows that there are still 692 complaints recorded in 2022, and statistics show that at least 80% of them are about price discrimination on e-commerce platforms such as Taobao, Jingdong and Meituan. If algorithmic price discrimination on e-commerce platforms is not effectively regulated, it will result in a winner-takes-all monopoly situation, leading consumers to question the fairness of e-commerce product prices, and once consumer trust is lost, it is difficult to regain, which will have irreversible negative effects [4,5].

Therefore, this paper aims to simulate the impact of different factors on the regulation of algorithmic price discrimination on e-commerce platforms using an evolutionary game approach. This is validated by simulation experiments, and, then, targeted countermeasures are proposed for the effective regulation of algorithmic price discrimination. This paper is of great practical significance for maintaining consumer fairness, increasing consumer trust and promoting the sustainable development of the e-commerce industry.

## 2. Literature Review

At present, many scholars have studied various aspects of algorithmic price discrimination on e-commerce platforms, including the mechanisms at play and behavioural regulation measures.

In terms of the mechanism of algorithmic price discrimination, the pricing of e-commerce products is an important factor affecting consumers' purchasing decisions, enterprises and platform revenues [6], and e-commerce platforms are typical market-creating bilateral markets with strong network externalities [7]. Moderate secondary and tertiary price discrimination can increase social welfare and achieve a win–win situation for all parties [8]. However, with the development of big data, algorithms and other technologies, firms are increasingly dependent on them when formulating price strategies [9]. Many experts and scholars believe that the algorithmic price discrimination practiced by e-commerce platforms is close to primary price discrimination [10], in which the algorithm can dynamically achieve "a thousand prices for a thousand people" [11] so as to maximize the payment price per unit of consumer demand while the platforms increase their own benefits and choose to ignore the social responsibility they should bear [12]. At this time, consumers will spend more time and energy to compare product prices, which will significantly affect consumers' purchase decisions [13]. Consumers have a preference for price fairness [14] and an aversion to privacy intrusion [15], and once they find that prices are unfair, a strong sense of betrayal will arise [16], and they will take action to complain and expand the influence of social opinion [17].

The regulation of the algorithmic price discrimination behaviour of e-commerce platforms is a dynamic and continuous process with multiple influencing factors [18]. Evolutionary games believe that the subject of the game is limited and rational and can effectively analyse the influencing factors in the decision-making process, continuously learn and improve the game strategy [19,20], and finally approach the equilibrium state [21]. They have been widely used in environmental protection [22,23], finance [24,25], medicine [26,27], management [28,29] and other fields. Evolutionary games are consistent with the actual process of behaviour regulation [30–32]. Some scholars have applied evolutionary games to the study of algorithmic price discrimination regulation measures: Wu Bin [33] constructs an evolutionary game model between the government and e-commerce platforms, arguing that increasing government penalties can effectively motivate the government to actively regulate and transform the algorithmic price discrimination strategy of e-commerce platforms; Pan Ding [34] argues that consumers are disadvantaged when experiencing algorithmic price discrimination, and, therefore, constructs an evolutionary game model between e-commerce enterprises and the government to consider the optimal strategy choice between traditional and big data regulation methods under the role of different contextual influence coefficients. However, e-commerce platforms have the advantage of asymmetric data information [35] and opaque algorithms [36]. Meanwhile, the government's single-party regulatory capacity is limited [37], makes it difficult to effectively

monitor and regulate their algorithmic price discrimination behaviours [38]. As actual purchasers of products and services, consumers are more price-sensitive [39], and consumers, as stakeholders, can use their social graph radiation to amplify perceived unfair pricing information and compete with algorithmic price discrimination [40]. Therefore, Xing Genshang [41] considers the evolutionary game between consumers and e-commerce platforms on both sides when consumers have the right to data portability and explores the regulation strategy of algorithmic price discrimination behaviour from the user's perspective.

In summary, the existing research takes the mechanism of algorithmic price discrimination on e-commerce platforms as the starting point, revealing its negative impact and affirming the need to regulate algorithmic price discrimination [42]. It uses evolutionary games to analyse the strategy choices of the participants in different situations and proposes targeted recommendations for measures to regulate algorithmic price discrimination [34,35,41].

However, on one hand, the practice has shown that the role of consumer co-regulation in the process of regulating algorithmic price discrimination cannot be ignored, and existing studies often only consider the game between the government and the platform, and the consumer and the platform, and only consider the third party as an influential factor in the game, ignoring the complex interests of the three parties in reality. On the other hand, due to the non-disclosure of platform algorithms, the concealment of dynamic algorithmic price discrimination, it is not always possible for government and consumer regulation to detect algorithmic price discrimination by the platforms [43]. Further consideration needs to be given to the probability of the government and consumers detecting algorithmic price discrimination at the time of regulation and the equilibrium strategy choices of the three parties involved in different contexts.

Therefore, this paper will construct a three-party evolutionary game model of e-commerce platforms, local governments and consumers. We comprehensively analyse the evolutionary stabilisation strategies of the three parties involved under the roles of the e-commerce platform's algorithmic price discrimination strength, the probability of government regulation discovering price discrimination and the consumers' price fairness preferences and collaborative regulation with the government. We verify these strategies numerically by substituting simulation experiments and analyse the influence of each factor on the equilibrium state of the system. Finally, the conclusions obtained are combined with recommendations for the regulation of price discrimination in e-commerce platforms.

The contribution and innovation of this paper is to consider the complex interests of e-commerce platforms, local governments and consumers and, to a certain extent, to broaden the research horizon of algorithmic price discrimination in e-commerce platforms. In addition, the probability of finding algorithmic price discrimination is also considered, making the study more relevant.

## 3. Basic Assumptions and Model Construction

### 3.1. Description of the Problem

The process of regulating algorithmic price discrimination on e-commerce platforms is complex. Squeezed consumers have no right to enforce the law and can only defend their rights with public opinion. The lower bargaining power often leads to switch channels of consumption. If only local governments were to regulate, on the one hand, the algorithmic black box of e-commerce platforms would make government regulation difficult, and the marginal utility of regulation would be low. On the other hand, algorithmic discrimination could significantly increase the economic revenue of e-commerce platforms, which in turn would, to a certain extent, drive local economic development and increase local government tax revenues. Without the introduction of third-party regulation, local governments may easily choose to collude with e-commerce platforms and turn a blind eye to their short-sighted use of algorithmic price discrimination. That will result in e-commerce platforms recklessly extracting residual value from consumers, causing consumers to lose trust in the platforms and local governments, undermining the credibility of the government and hindering the sustainable development of the e-commerce industry.

Therefore, this paper treats the finite rational e-commerce platform, the local government and the consumers as the three parties involved in the game. Based on the parametric assumptions, a payoff matrix is built, and the replicated dynamic system is constructed and solved according to the expected payoffs. The evolutionary equilibrium state strategy is studied as it continuously learns and improves its own strategy. The game relationship between the participating subjects is shown in Figure 1 below.

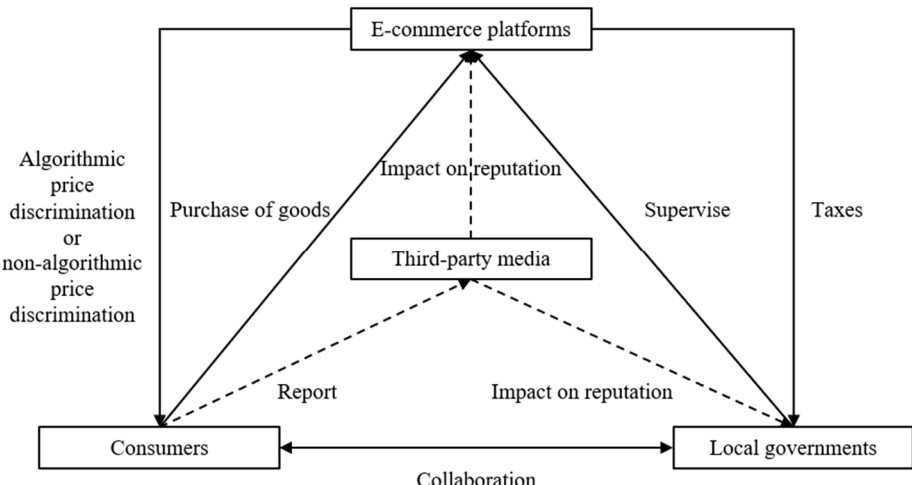

**Figure 1.** Diagram of the game relationship between the participating players.

*3.2. Basic Assumptions*

In order to analyse the strategic choices of the e-commerce platforms, the local governments and the consumers in the evolutionary game, the following assumptions are made to ensure that the nature of the actual problem remains unchanged:

**Hypothesis 1.** *Assume that the e-commerce platform (P) has a strategy space of (algorithmic price discrimination, non-algorithmic price discrimination), for which the corresponding choice probability is $(x, 1 - x)$ and $0 < x < 1$. Let the normal revenue of the e-commerce platform when it chooses not to use algorithmic price discrimination be $\Delta R_P$ and the additional revenue gained by using algorithmic price discrimination to extract the residual value of consumers be $\Delta R_P$. The daily operation and maintenance cost of the platform is $C_{P1}$, and the additional technical cost of algorithmic price discrimination is $C_{P2}$. When algorithmic price discrimination by an e-commerce platform is revealed, it leads to a variety of losses $L_{P1}$ such as reduced reputation, weakened network externalities for loyal consumers and a reduced horse-trading effect for new users.*

**Hypothesis 2.** *Assume that the strategy space of the local government (G) is (positive regulation, negative regulation) and that its corresponding choice probability is $(y, 1 - y)$ and $0 < y < 1$. Let the tax rate levied by the local government for the e-commerce platform be g, while its tax is positively related to the revenue of the e-commerce platform. When the local government chooses to actively regulate the cost of technology and manpower, it is represented as $C_{G1}$. If the local government conspires with the e-commerce platform and chooses to negatively regulate its algorithmic price discrimination, it will bear the burden of declining public trust and accountability penalties from the central government total $L_{G1}$. When consumers discover that the platform is practicing algorithmic price discrimination and choose a collaborative regulatory strategy to actively report it to the local government, if the local government chooses a negative regulatory strategy, consumers will feel unjust because they have paid the time and energy costs but have no way to complain and have not allowed the platform to receive its due, which will lead to a decline in the credibility of the local government $L_{G2}$. Conversely, the reputation of local governments will be enhanced when they collaborate with consumers to regulate algorithmic discrimination $R_G$; if local governments find that e-commerce platforms are using algorithmic price discrimination in their regulation, they*

*will be penalised with fines F and will be required to pay compensation to consumers for the price difference D.*

**Hypothesis 3.** *Assume that the strategy space for consumers ( C) is (positive regulation, negative regulation) and their corresponding choice probability is $(z, 1 - z)$, and $0 < z < 1$. Let the consumer's expected price for the product sold by the e-commerce platform be $P_C$. When the platform does not carry out algorithmic price discrimination, the price of the product is $P_1$ because the platform cannot accurately obtain the expected price of consumers, and, in order to expand product sales, generally $P_1 < P_C$, the benefits to consumers when the platform does not practice algorithmic price discrimination are $R_{C1}$ ($R_{C1} = P_C - P_1$). Alternatively, if the platform implements algorithmic price discrimination, with the help of algorithms, big data technology will have the ability to achieve a thousand different prices for a thousand different people. At this point, the product price converges to $P_C$; that is, when the platform implements algorithmic price discrimination, the consumer's benefit $R_C$ tends to 0. If consumers choose to actively regulate the platform, they will collect evidence and report to the local government if they find algorithmic price discrimination, which will cost them time and effort $C_{C1}$, and they will receive a bonus S from the local government for actively cooperating with the platform.*

**Hypothesis 4.** *Let the degree of algorithmic price discrimination implemented by the platform be $\varphi$ ($\varphi = \frac{P_C - P_1}{P_1} = \frac{P_C}{P_1} - 1$, the implementation of algorithmic price discrimination product premium to the normal price ratio). In order to be realistic and not to lose generality, the additional revenue, additional technical costs, losses, fines and price difference compensation from the platform algorithmic price discrimination are positively related to the degree of algorithmic price discrimination. When algorithmic price discrimination is implemented on a platform, let the consumer's price fairness preference be $\alpha$. A higher preference means that the more concerned about price fairness, the greater the likelihood of detecting algorithmic price discrimination when actively regulated. Let the probability of detecting algorithmic price discrimination when actively regulated by the government be $\beta$. When consumers and the government collaborate in regulation, the probability of being able to effectively detect algorithmic price discrimination on a platform is 1 ( $\alpha < 1$, $\beta < 1$).*

*3.3. Model Construction*

Based on the above assumptions, the tripartite evolutionary game model payoff matrix is constructed as shown in Table 1:

**Table 1.** Matrix of payoffs for the three-party evolutionary game.

| E-Commerce Platforms | Local Governments | Consumers | |
|---|---|---|---|
| | | **Active Regulation Z** | **Negative Regulation $1 - Z$** |
| Algorithmic price discrimination $x$ | Active regulation $y$ | $(1-g)(R_{P1} + \varphi\Delta R_P) - C_{P1} - \varphi C_{P2}$ $-\varphi(F+D) - L_{P1}$ $g(R_{P1} + \varphi\Delta R_P) - C_{G1} + \varphi F - S + R_G$ $D - C_{C1} + S$ | $(1-g)(R_{P1} + \varphi\Delta R_P) - C_{P1} - \varphi C_{P2}$ $-\beta[\varphi(F+D) + L_{P1}]$ $g(R_{P1} + \varphi\Delta R_P) - C_{G1} + \beta\varphi F$ $\beta D$ |
| | Negative regulation $1-y$ | $(1-g)(R_{P1} + \varphi\Delta R_P) - C_{P1} -$ $\varphi C_{P2} - \alpha L_{P1}$ $g(R_{P1} + \varphi\Delta R_P) - L_{G1} - \alpha L_{G2}$ $-C_{C1}$ | $(1-g)(R_{P1} + \varphi\Delta R_P) - C_{P1} - \varphi C_{P2}$ $g(R_{P1} + \varphi\Delta R_P) - L_{G1}$ $0$ |
| Non-algorithmic price discrimination $1-x$ | Active regulation $y$ | $(1-g)R_{P1} - C_{P1}$ $gR_{P1} - C_{G1} - S + R_G$ $R_{C1} - C_{C1} + S$ | $(1-g)R_{P1} - C_{P1}$ $gR_{P1} - C_{G1}$ $R_{C1}$ |
| | Negative regulation $1-y$ | $(1-g)R_{P1} - C_{P1}$ $gR_{P1} - L_{G1}$ $R_{C1} - C_{C1}$ | $(1-g)R_{P1} - C_{P1}$ $gR_{P1} - L_{G1}$ $R_{C1}$ |

From the matrix of payment benefits, the following can be drawn:

The expected benefits of algorithmic price discrimination $U_{P1}$, the expected benefits of non-algorithmic price discrimination $U_{P2}$ and the average expected benefits of the e-commerce platforms $\overline{U_P}$ are as follows:

$$
\begin{aligned}
U_{P1} =\ & yz((1-g)(R_{P1} + \varphi\Delta R_P) - C_{P1} - \varphi C_{P2} - \varphi(F+D) - L_{P1}) \\
& +y(1-z)((1-g)(R_{P1} + \varphi\Delta R_P) - C_{P1} - \varphi C_{P2} - \beta(\varphi(F+D) + L_{P1})) \\
& +(1-y)z((1-g)(R_{P1} + \varphi\Delta R_P) - C_{P1} - \varphi C_{P2} - \alpha L_{P1}) \\
& +(1-y)(1-z)((1-g)(R_{P1} + \varphi\Delta R_P) - C_{P1} - \varphi C_{P2})
\end{aligned}
\tag{1}
$$

$$
\begin{aligned}
U_{P2} =\ & yz((1-g)R_{P1} - C_{P1}) + y(1-z)((1-g)R_{P1} - C_{P1}) \\
& +(1-y)z((1-g)R_{P1} - C_{P1}) + (1-y)(1-z)((1-g)R_{P1} - C_{P1})
\end{aligned}
\tag{2}
$$

$$
\overline{U_P} = x U_{P1} + (1-x)U_{P2}
\tag{3}
$$

The replication dynamics equation for e-commerce platforms is

$$
\begin{aligned}
F_P(x) = \frac{dx}{dt} =\ & x(1-x)\{y[z((\alpha + \beta - 1)L_{P1} + (\beta - 1)\varphi(F+D)) - \beta(\varphi(F+D) + L_{P1})] \\
& -z\alpha L_{P1} + (1-g)\varphi\Delta R_P - \varphi C_{P2}\}
\end{aligned}
\tag{4}
$$

The expected benefits of active regulation $U_{G1}$, the expected benefits of passive regulation $U_{G2}$ and the average expected benefits for local governments $\overline{U_G}$ are as follows:

$$
\begin{aligned}
U_{G1} =\ & xz(g(R_{P1} + \varphi\Delta R_P) - C_{G1} + \varphi F + R_G - S) \\
& +x(1-z)(g(R_{P1} + \varphi\Delta R_P) - C_{G1} + \beta\varphi F) \\
& +(1-x)z(gR_{P1} - C_{G1} - S + R_G) + (1-x)(1-z)(gR_{P1} - C_{G1})
\end{aligned}
\tag{5}
$$

$$
\begin{aligned}
U_{G2} =\ & xz(g(R_{P1} + \varphi\Delta R_P) - L_{G1} - \alpha L_{G2}) + x(1-z)(g(R_{P1} + \varphi\Delta R_P) - L_{G1}) \\
& +(1-x)z(gR_{P1} - L_{G1}) + (1-x)(1-z)(gR_{P1} - L_{G1})
\end{aligned}
\tag{6}
$$

$$
\overline{U_G} = y U_{G1} + (1-y)U_{G2}
\tag{7}
$$

The replication dynamics equation for local governments is

$$
F_G(y) = \frac{dy}{dt} = y(1-y)\{x[z(\varphi F - \beta\varphi F + \alpha L_{G2}) + \beta\varphi F] + z(R_G - S) + L_{G1} - C_{G1}\}
\tag{8}
$$

The expected benefits of consumers' choice of regulation $U_{C1}$, the expected benefits of negative regulation $U_{C2}$ and the average expected benefits $\overline{U_C}$ are as follows:

$$
\begin{aligned}
U_{C1} =\ & xy(\varphi D - C_{C1} + S) - x(1-y)C_{C1} \\
& +(1-x)y(R_{C1} - C_{C1} + S) + (1-x)(1-y)(R_{C1} - C_{C1})
\end{aligned}
\tag{9}
$$

$$
U_{C2} = xy\beta\varphi D + (1-x)yR_{C1} + (1-x)(1-y)R_{C1}
\tag{10}
$$

$$
\overline{U_C} = z U_{C1} + (1-z)U_{C2}
\tag{11}
$$

The replication dynamics equation for consumers is

$$
F_C(z) = \frac{dz}{dt} = z(1-z)\{x[y\varphi D(1-\beta)] + yS - C_{C1}\}
\tag{12}
$$

## 4. Analysis of System Evolutionary Stabilisation Strategies

According to the stability theorem of the differential equation, if $\frac{dF_P(x)}{dx} < 0$, $\frac{dF_G(y)}{dy} < 0$, $\frac{dF_C(z)}{dz} < 0$, it means that the e-commerce platforms, local governments and consumers tend to stabilise their strategic choices and develop the analysis as follows:

### 4.1. Analysis of the Stability of E-Commerce Platform Strategies

A first order derivative of the replication dynamics equation for the e-commerce platforms gives the following equation:

$$\frac{dF_P(x)}{dx} = (1-2x)\{y[z((\alpha+\beta-1)L_{P1}+(\beta-1)(\varphi F+\varphi D))-\beta(\varphi(F+D)+L_{P1})]$$
$$-z\alpha L_{P1}+(1-g)\varphi \Delta R_P-\varphi C_{P2}\} \tag{13}$$

Conclusion 1: When $y = \frac{z\alpha L_{P1}-(1-g)\varphi \Delta R_P+\varphi C_{P2}}{z[(\alpha+\beta-1)L_{P1}+(\beta-1)(\varphi F+\varphi D)]-\beta(\varphi F+L_{P1}+\varphi D)}$ (for convenience, it is written it as $\phi_1$), if $0 < \phi_1 < 1$ and $\frac{dF_P(x)}{dx} \equiv 0$, the e-commerce platforms are in a stable strategy state regardless of the value of $x$.

Conclusion 2: When $0 < y < \phi_1 < 1$, $\frac{dF_P(x)}{dx}|_{x=0} < 0$ and $\frac{dF_P(x)}{dx}|_{x=1} > 0$, then $x = 0$ is the stable strategy state. This means that e-commerce platforms will not engage in algorithmic price discrimination when the probability of local governments choosing to regulate is below the threshold $\phi_1$.

Conclusion 3: When $0 < \phi_1 < y < 1$, $\frac{dF_P(x)}{dx}|_{x=0} > 0$ and $\frac{dF_P(x)}{dx}|_{x=1} < 0$, then $x = 1$ is the stable strategy state. This means that e-commerce platforms will engage in algorithmic price discrimination when the probability of local governments choosing to regulate is above the threshold $\phi_1$. The phase diagram of the spatial evolution of e-commerce platforms' strategies is shown in Figure 2.

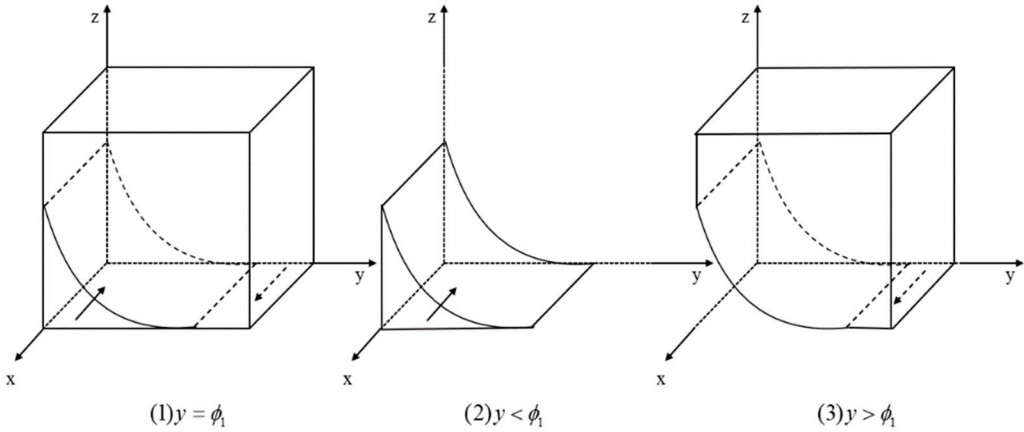

**Figure 2.** Phase diagram of the strategic evolution of e-commerce platforms.

### 4.2. Analysis of the Stability of Local Government Strategies

A first order derivative of the replication dynamic equation for local governments gives the following equation:

$$\frac{dF_G(y)}{dy} = (1-2y)\{x[z(\varphi F-\beta\varphi F+\alpha L_{G2})+\beta\varphi F]+z(R_G-S)+L_{G1}-C_{G1}\} \tag{14}$$

Conclusion 1: When $x = \frac{z(R_G-S)-L_{G1}+C_{G1}}{z(\varphi F-\beta\varphi F+\alpha L_{G2})+\beta\varphi F}$ (for convenience, it is written as $\phi_2$), if $0 < \phi_2 < 1$ and $\frac{dF_G(y)}{dy} \equiv 0$, the local governments are in a stable strategy state regardless of the value of $y$.

Conclusion 2: When $0 < x < \phi_2 < 1$, $\frac{dF_G(y)}{dy}|_{y=0} < 0$ and $\frac{dF_G(y)}{dy}|_{y=1} > 0$, then $y = 0$ is the stable strategy state. This means that local governments will adopt a negative regulatory strategy when the probability of e-commerce platforms choosing to price discriminate is below the threshold $\phi_2$.

Conclusion 3: When $0 < \phi_2 < x < 1$, $\frac{dF_G(y)}{dy}|_{y=0} > 0$ and $\frac{dF_G(y)}{dy}|_{y=1} < 0$, then $y = 1$ is the stable strategy state. This means that local governments adopt an active regulatory strategy when the probability of e-commerce platforms choosing to price discriminate is

above the threshold $\phi_2$. The phase diagram of the spatial evolution of local governments' strategies is shown in Figure 3.

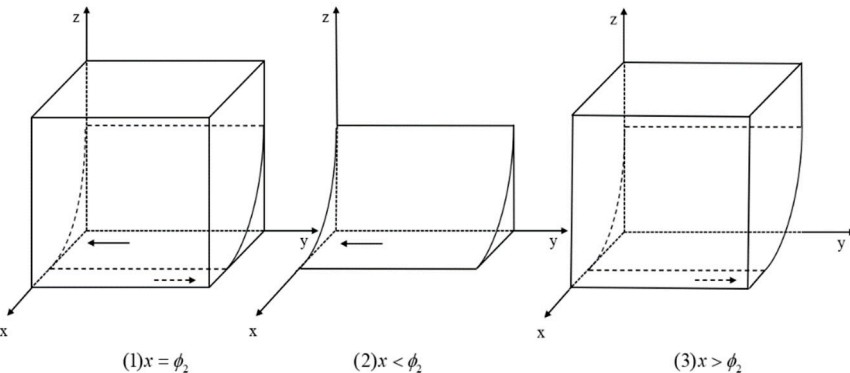

(1)$x = \phi_2$    (2)$x < \phi_2$    (3)$x > \phi_2$

**Figure 3.** Phase diagram of the strategic evolution of local governments.

*4.3. Analysis of the Stability of Consumer Strategies*

A first order derivative of the replication dynamic equation for consumers gives the following equation:

$$\frac{dF_C(z)}{dz} = (1 - 2z)\{x[yD(1 - \beta)] + yS - C_{C1}\} \tag{15}$$

Conclusion 1: When $x = \frac{C_{C1} - yS}{y\varphi D(1 - \beta)}$ (for convenience, it is written as $\phi_3$), if $0 < \phi_3 < 1$ and $\frac{dF_C(z)}{dz} \equiv 0$, the consumers are in a stable strategy state regardless of the value of $z$.

Conclusion 2: When $0 < x < \phi_3 < 1$, $\frac{dF_C(z)}{dz}|_{z=0} < 0$ and $\frac{dF_C(z)}{dz}|_{z=1} > 0$, then $z = 0$ is the stable strategy state. This means that consumers will adopt a negative regulatory strategy when the probability of e-commerce platforms choosing to price discriminate is below the threshold $\phi_3$.

Conclusion 3: When $0 < \phi_3 < x < 1$, $\frac{dF_C(z)}{dz}|_{z=0} > 0$ and $\frac{dF_C(z)}{dz}|_{z=1} < 0$, then $z = 1$ is the stable strategy state, This means that consumers will adopt an active regulatory strategy when the probability of e-commerce platforms choosing to price discriminate is above the threshold $\phi_3$. The phase diagram of the spatial evolution of consumers' strategies is shown in Figure 4.

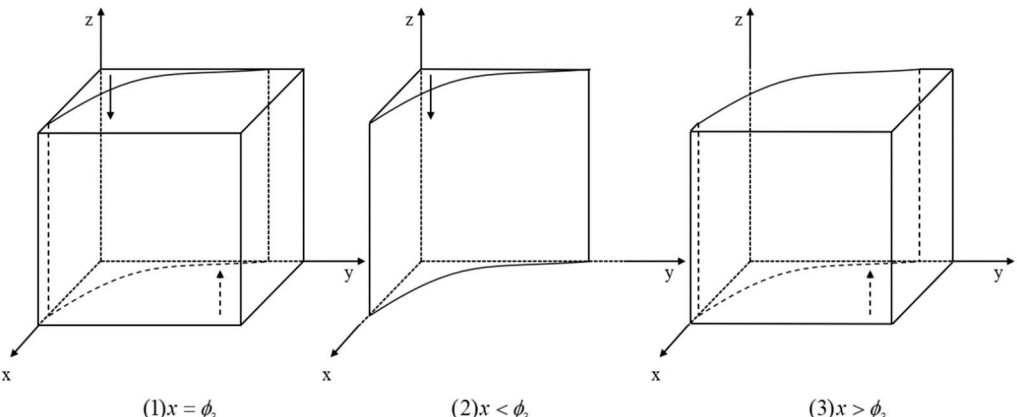

(1)$x = \phi_3$    (2)$x < \phi_3$    (3)$x > \phi_3$

**Figure 4.** Phase diagram of the strategic evolution of consumers.

*4.4. Stability Analysis of Tripartite Evolutionary Game System*

The simultaneous Formulas (4), (8) and (12) form a system of replication dynamic equations, and when $F_P(x), F_G(y), F_C(z)$ is 0, there are 8 local stability points for pure

strategies and 1 local equilibrium point for mixed strategies in the three-dimensional interval of $S= \{(x,y,z)|0 \leq x \leq 1, 0 \leq y \leq 1, 0 \leq z \leq 1\}$, which include $E_1(0,0,0)$, $E_2(0,1,0)$, $E_3(0,0,1)$, $E_4(0,1,1)$, $E_5(1,0,0)$, $E_6(1,1,0)$, $E_7(1,0,1)$, $E_8(1,1,1)$ and $E_9(\phi_1, \phi_2, \phi_3)$.

The partial derivatives of $x$, $y$, $z$ for the replicated dynamic equations and the Jacobian matrix are obtained as follows:

$$
J = \begin{bmatrix} \dfrac{\partial F_P(x)}{\partial x} & \dfrac{\partial F_P(x)}{\partial y} & \dfrac{\partial F_P(x)}{\partial z} \\[2mm] \dfrac{\partial F_G(y)}{\partial x} & \dfrac{\partial F_G(y)}{\partial y} & \dfrac{\partial F_G(y)}{\partial z} \\[2mm] \dfrac{\partial F_C(z)}{\partial x} & \dfrac{\partial F_C(z)}{\partial y} & \dfrac{\partial F_C(z)}{\partial z} \end{bmatrix} \tag{16}
$$

Among them:

$\frac{\partial F_P(x)}{\partial x} = (1-2x)\{y[z((\alpha+\beta-1)L_{P1} + (\beta-1)(\varphi F+\varphi D))- \beta(\varphi F + L_{P1} + \varphi D)] - z\alpha L_{P1} + (1-g)\varphi\Delta R_P - \varphi C_{P2}\}'$

$\frac{\partial F_P(x)}{\partial y} = x(1-x)[z((\alpha+\beta-1)L_{P1} + (\beta-1)(\varphi F+\varphi D)) - \beta(\varphi F + L_{P1} + \varphi D)];$

$\frac{\partial F_P(x)}{\partial z} = x(1-x)\{y[(\alpha+\beta-1)L_{P1} + (\beta-1)(\varphi F+\varphi D)] - \alpha L_{P1}\};$

$\frac{\partial F_G(y)}{\partial x} = y(1-y)[z(\varphi F - \beta\varphi F + \alpha L_{G2}) + \beta\varphi F];$

$\frac{\partial F_G(y)}{\partial y} = (1-2y)\{x[z(\varphi F - \beta\varphi F + \alpha L_{G2}) + \beta\varphi F] + z(R_G - S) + L_{G1} - C_{G1}\};$

$\frac{\partial F_G(y)}{\partial z} = y(1-y)\{x(\varphi F - \beta\varphi F + \alpha L_{G2}) + R_G - S\};$

$\frac{\partial F_C(z)}{\partial x} = z(1-z)[y\varphi D(1-\beta)];$

$\frac{\partial F_C(z)}{\partial y} = z(1-z)[x\varphi D(1-\beta) + S];$

$\frac{\partial F_C(z)}{\partial z} = (1-2z)\{x[y\varphi D(1-\beta)] + yS - C_{C1}\}.$

Since $E_9(\phi_1, \phi_2, \phi_3)$ makes $Tr(J) = 0$, which is the centre point, it does not reach a steady state, and only the stability of the 8 pure strategy local equilibrium points is discussed. According to the Lyapunov first law, if all $\lambda < 0$ in the corresponding points of the Jacobian matrix, the point is the stable equilibrium point of the system. If all $\lambda > 0$, the point is unstable. If there is both $\lambda < 0$ and $\lambda > 0$, it is a saddle point. The calculated characteristic values of each point are shown in Table 2.

**Table 2.** Jacobian matrix eigenvalues of local equilibrium points of pure strategies.

| Local Equilibrium Point | $\lambda_1$ | $\lambda_2$ | $\lambda_3$ |
|---|---|---|---|
| $E_1(0,0,0)$ | $(1-g)\varphi\Delta R_P - \varphi C_{P2}$ | $L_{G1} - C_{G1}$ | $-C_{C1}$ |
| $E_2(0,1,0)$ | $(1-g)\varphi\Delta R_P - \varphi C_{P2} - \beta(\varphi F + \varphi D + L_{P1})$ | $C_{G1} - L_{G1}$ | $S - C_{C1}$ |
| $E_3(0,0,1)$ | $(1-g)\varphi\Delta R_P - \varphi C_{P2} - \alpha L_{P1}$ | $L_{G1} - C_{G1} - S + R_g$ | $C_{C1}$ |
| $E_4(0,1,1)$ | $(1-g)\varphi\Delta R_P - \varphi C_{P2} - (\varphi F + \varphi D + L_{P1})$ | $C_{G1} - L_{G1} + S - R_g$ | $C_{C1} - S$ |
| $E_5(1,0,0)$ | $\varphi C_{P2} - (1-g)\varphi\Delta R_P$ | $L_{G1} - C_{G1} + \beta\varphi F$ | $-C_{C1}$ |
| $E_6(1,1,0)$ | $\varphi C_{P2} - (1-g)\varphi\Delta R_P + \beta(\varphi F + \varphi D + L_{P1})$ | $C_{G1} - L_{G1} - \beta\varphi F$ | $(1-\beta)\varphi D - C_{C1} + S$ |
| $E_7(1,0,1)$ | $\varphi C_{P2} - (1-g)\varphi\Delta R_P + \alpha L_{P1}$ | $L_{G1} - C_{G1} + \alpha L_{G2} + \varphi F - S + R_g$ | $C_{C1}$ |
| $E_8(1,1,1)$ | $\varphi C_{P2} - (1-g)\varphi\Delta R_P + \varphi F + \varphi D + L_{P1}$ | $C_{G1} - L_{G1} - \alpha L_{G2} - \varphi F + S - R_g$ | $C_{C1} - (1-\beta)\varphi D - S$ |

Since the corresponding scenario of the local equilibrium point feature value of the pure strategy is more complex, in order to facilitate analysis and be as close to the actual life situation as possible, and without considering the impact of government and consumer punishment measures, when the e-commerce platform carries out algorithmic price discrimination, the additional benefits obtained far exceed the benefits when there is no algorithmic price discrimination, which is represented as $(1-g)\varphi\Delta R_P - \varphi C_{P2} > 0$. The cost of active supervision by local governments is less than the penalties imposed by negative supervision, which is represented as $L_{G1} - C_{G1} > 0$; Moreover, the bonus given

by local governments to consumers for active collaborative supervision is less than the reputation-enhancing benefits brought by cooperation, which is represented as $S - R_g < 0$.

When $(1 - g)\varphi\Delta R_P - \varphi C_{P2} < \beta(\varphi F + \varphi D + L_{P1})$, when the excess grey income obtained by the e-commerce platform from algorithmic price discrimination is less than the sum of the fines levied by active supervision and the consumer network externalities lost by the platform, it can effectively promote the reasonable pricing of the e-commerce platform, which can be divided into the following two situations:

Situation 1: When $S < C_{C1}$, the collaborative supervision bonus given to consumers by the local government cannot easily cover its regulatory costs, and the characteristic values of $E_2(0, 1, 0)$, which is the equilibrium point of the gradual evolution of the system, are negative. At this time, the e-commerce platform chooses the non-algorithmic price discrimination strategy, the local government chooses the active regulatory strategy and the consumers choose the negative regulatory strategy.

Situation 2: When $S > C_{C1}$, the cost of consumer collaborative supervision is less than the bonus given by the local government. The characteristic values of $E_4(0, 1, 1)$, which is the equilibrium point of the gradual evolution of the system, are negative, and the bonus is higher than the regulatory cost, which gives consumers the motivation to actively supervise. At this time, the e-commerce platform chooses the non-algorithmic price discrimination strategy, the local government chooses the active supervision strategy and the consumers choose the active supervision strategy.

When $(1 - g)\varphi\Delta R_P - \varphi C_{P2} > \beta(\varphi F + \varphi D + L_{P1})$, the excess grey income obtained by the e-commerce platform from algorithm price discrimination is greater than the sum of the fines levied by active supervision and the consumer network externalities lost by the platform. It is difficult to effectively regulate the algorithm price discrimination behaviour of the e-commerce platform, which can be divided into the following two situations:

Situation 3: When $C_{C1} > (1 - \beta)\varphi D + S$, the cost of active supervision by consumers is greater than the amount of additional price difference compensation brought by active supervision and the bonus given by the local government. The characteristic values of $E_6(1, 1, 0)$, which is the equilibrium point of gradual evolution of the system, are negative, at which time the e-commerce platform chooses the algorithm price discrimination strategy, the local government chooses the active supervision strategy and the consumer chooses the negative supervision strategy.

Situation 4: When $C_{C1} < (1 - \beta)\varphi D + S$, the amount of additional price difference compensation brought by consumers' active supervision and the bonus given by the local government are greater than the cost consumed by active supervision. The characteristic values of $E_8(1, 1, 1)$, which is the equilibrium point of gradual evolution of the system, are negative, at which time the e-commerce platform chooses the algorithm price discrimination strategy, the local government chooses the active supervision strategy and the consumer chooses the active supervision strategy.

## 5. Numerical Simulation and Analysis of Influencing Factors

### 5.1. Numerical Simulation

In order to reflect the evolution process of the system more intuitively and accurately, numerical simulation experiments were carried out using Matlab R2021 b. Based on the actual situation, the tax rate paid by the e-commerce platform is $g = 0.06$; according to the results of the Beijing Consumers Association's questionnaire research results, the degree of price discrimination of the e-commerce platform is $\varphi = 0.85$; and, referring to Pan Ding [15], when the local government actively supervises, the probability of finding the algorithm price discrimination of the e-commerce platform is $\beta = 0.8$. Combined with the restrictions under different circumstances, set the remaining parameter values as shown in the Table 3 below.

To substitute the parameter values into the three-dimensional replication dynamic system, set the initial willingness loop function as $for \ x(y, z) = 0:0.2:1$ for participating tripartite subjects. The experimental results of the numerical simulation for scenarios

1–4 above are obtained as shown in Figure 5a–d. Where the different coloured lines represent the strategy choices of the participating subjects in different initial states. These results validate the inference of asymptotic stability points for the evolution of the system for different scenarios, and, as can be seen from the figure, the initial willingness to choose a strategy by the participating tripartite subjects is positively related to the rate of system evolution to an asymptotically stable state but cannot influence the final system stability outcome. Figure 5b is the most ideal state, in which local governments and consumers actively cooperate in supervision, and the excess grey income from the algorithm price discrimination of e-commerce platforms is less than the sum of regulatory fines and network externalities; that is, the regulatory punishment measures can effectively promote e-commerce platforms to choose non-algorithm price discrimination strategies. Figure 5d shows the least ideal state, in which the regulatory penalties are small, and it is difficult to cover the excess grey income of e-commerce platforms. Additionally, the local governments and consumers actively cooperate with supervision, but the e-commerce platforms still choose to carry out the short-sighted behaviour of algorithm price discrimination in order to obtain huge profits.

**Table 3.** Setting parameter values under different circumstances.

| Scenario | $\Delta R_P$ | $L_{P1}$ | $C_{P2}$ | $F$ | $C_{G1}$ | $L_{G1}$ | $L_{G2}$ | $R_G$ | $\alpha$ | $S$ | $C_{C1}$ | $D$ |
|---|---|---|---|---|---|---|---|---|---|---|---|---|
| 1 | 100 | 30 | 20 | 30 | 20 | 45 | 20 | 15 | 0.5 | 10 | 18 | 30 |
| 2 | 100 | 30 | 20 | 30 | 20 | 45 | 20 | 15 | 0.5 | 10 | 8 | 30 |
| 3 | 100 | 30 | 20 | 30 | 20 | 45 | 20 | 15 | 0.5 | 10 | 18 | 5 |
| 4 | 100 | 30 | 20 | 30 | 20 | 45 | 20 | 15 | 0.5 | 10 | 8 | 5 |

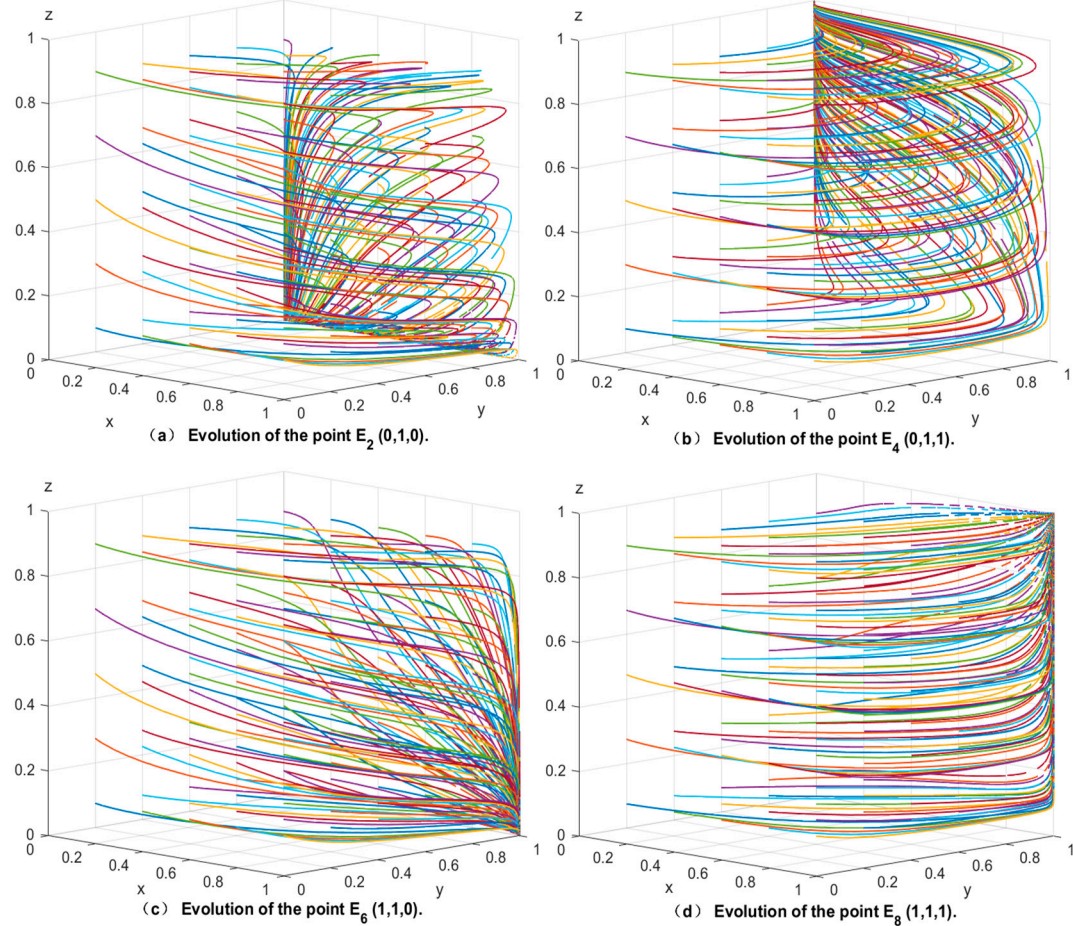

**Figure 5.** Numerical simulation in four scenarios.

*5.2. Sensitivity Analysis of Different Influencing Factors for System Evolution*

Based on the ideal scenario 2 condition, in order to avoid the influence of the initial intention of the three participants on the system evolution rate, the initial intention is selected as $(0.5, 0.5, 0.5)$. The analysis of the sensitivity of each influencing factor to the evolution of the system with an asymptotic steady state is described below.

5.2.1. Sensitivity Analysis of the Degree of Algorithm Price Discrimination of E-Commerce Platform for System Evolution

Let the values of the remaining influencing factors be fixed, and consider the degree of the price discrimination of the e-commerce platform algorithm, $\varphi = 0.75, 0.85, 0.95$. The system evolution result is shown in Figure 6 below. As can be seen from the graph, as the degree of algorithmic price discrimination on the e-commerce platforms deepens, the rate at which platforms choose not to engage in algorithmic price discrimination gradually slows down, while the rate at which local governments and consumers choose active regulatory strategies accelerates in unison. This shows that when the e-commerce platforms use algorithms to increase the degree of price discrimination, they can continuously squeeze the surplus value of consumers and obtain higher excess grey returns. However, at this time, consumers will be more active in coordinating supervision to protect themselves, while local governments will be more active to increase their credibility, and the punishments for the platforms will also increase. Thus, although the rate of convergence of e-commerce platforms slows down, eventually, non-algorithmic price discrimination will still be chosen.

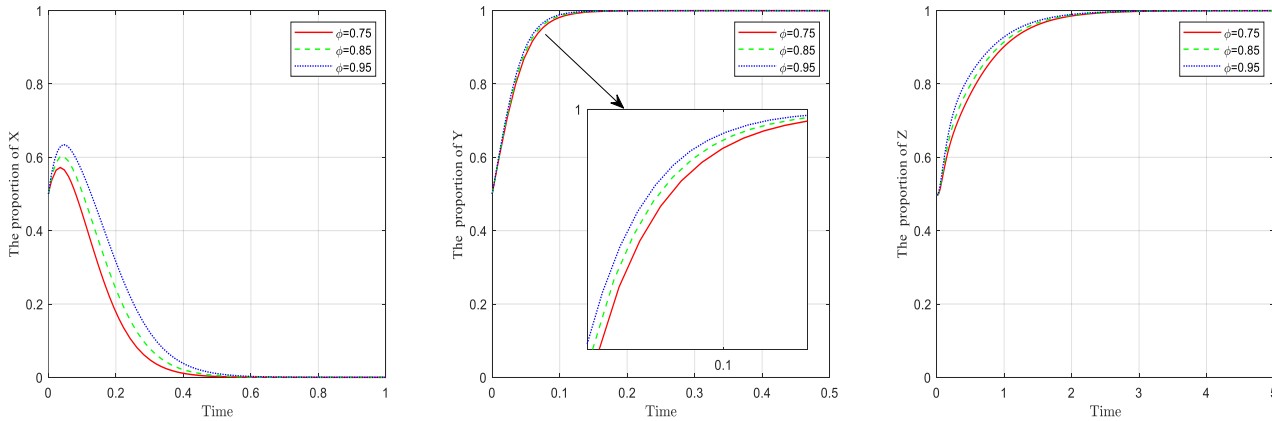

**Figure 6.** The effect of $\varphi$ on the evolution of the system to an asymptotic steady state.

5.2.2. Sensitivity Analysis of Excess Grey Returns from Algorithmic Price Discrimination on E-Commerce Platforms for System Evolution

Let the values of the remaining influencing factors be fixed and the excess grey income obtained by the algorithm price discrimination of the e-commerce platform be $\Delta R_P = 50, 100, 150$. The system evolution result is shown in Figure 7 below. It can be seen from this figure that, when $\Delta R_P$ is low, the e-commerce platform quickly converges to 0 and chooses a non-algorithmic price discrimination strategy but that the convergence speed slows down with the increase in $\Delta R_P$. When the excess grey return exceeds a certain threshold, the platform will choose algorithmic price discrimination, and the rate of convergence between local governments and consumers to active supervision will increase with the increase in $\Delta R_P$. It shows that the excess grey income obtained by the algorithm price discrimination of the e-commerce platform will significantly affect its strategy choice. When $\Delta R_P$ exceeds a certain threshold, the short-sighted behaviour of algorithm price discrimination will be carried out. At this time, it will evolve to the most undesirable situation of $(1, 1, 1)$. This, in turn, jeopardizes public rights and interests, affects its reputation, undermines the credibility of the government and causes adverse effects. While compelling local governments to increase punishments, consumers use third-party

media to expand their influence, use public opinion to amplify network externalities and effectively regulate e-commerce platforms' algorithm price discrimination.

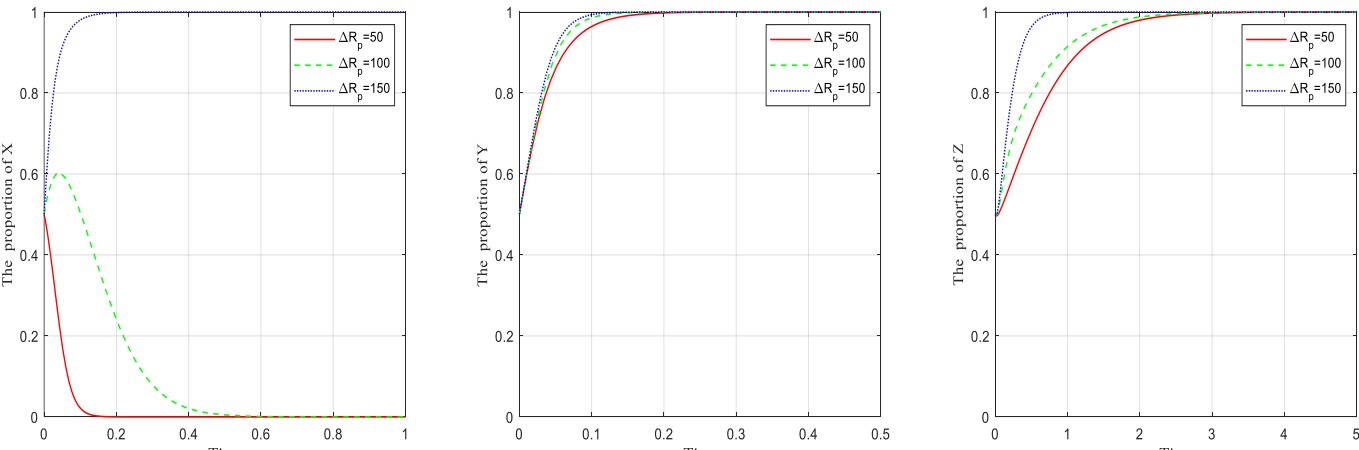

**Figure 7.** The effect of $\Delta R_P$ on the evolution of the system to an asymptotic steady state.

### 5.2.3. Sensitivity Analysis of the Probability of Algorithmic Price Discrimination When Local Governments Actively Supervise for System Evolution

Let the values of the remaining influencing factor parameters be fixed and the probability of the actively monitoring local government finding algorithmic price discrimination be $\beta = 0.4, 0.6, 0.8$. The system evolution result is shown in Figure 8 below. It can be seen from the figure that, when $\beta$ is at a low level, the willingness of platforms to carry out algorithmic price discrimination is stronger at first. Local governments and consumers actively supervise at this time, which prompts e-commerce platforms to turn to non-algorithmic price discrimination strategies. Additionally, with the increase in $\beta$, the rate of convergence to non-algorithmic price discrimination of platforms gradually increases. The rate of convergence to active supervision by local governments is basically unchanged. In turn, the rate of convergence to active regulation by consumers, out of trust in the government's ability to regulate and because of a free-rider mentality, gradually decreases, reaching an ideal equilibrium in which the public interest is safeguarded.

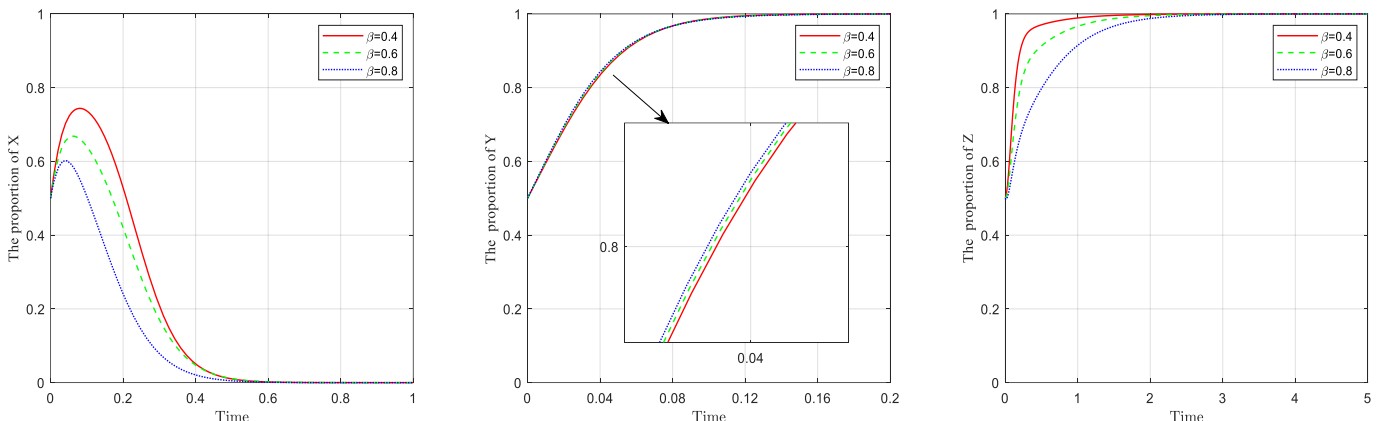

**Figure 8.** The effect of $\beta$ on the evolution of the system to an asymptotic steady state.

### 5.2.4. Sensitivity Analysis of Local Government Fines for Platform Algorithm Price Discrimination for System Evolution

Let the values of the remaining influencing factors be fixed and the penalty of local governments for price discrimination against platform algorithms be $F = 5, 30, 55$. The system evolution result is shown in Figure 9 below. It can be seen from the figure that, when $F$ is lower than a certain threshold, the lower penalty will give the platform the

opportunity to carry out the short-sighted behaviour of algorithm price discrimination. At this time, local governments and consumers will actively coordinate supervision, and, due to the damage to consumers' own rights and interests, their willingness to supervise is relatively stronger. With the increase in $F$, when the penalty breaks the threshold of the platform's algorithmic price discrimination balance, the platform will converge to a non-algorithmic price discrimination strategy. Additionally, the larger the value of $F$, the faster the convergence rate. Local governments are gaining more economic benefits from increased fines, so the rate of convergence to active regulatory strategies is increasing. Although consumers will still choose active supervision out of the psychology of free riding, their convergence rate will slow down. At this time, by increasing the amount of compensation, the sharing price data and information to reduce the cost of public supervision, the enthusiasm of consumers will also increase.

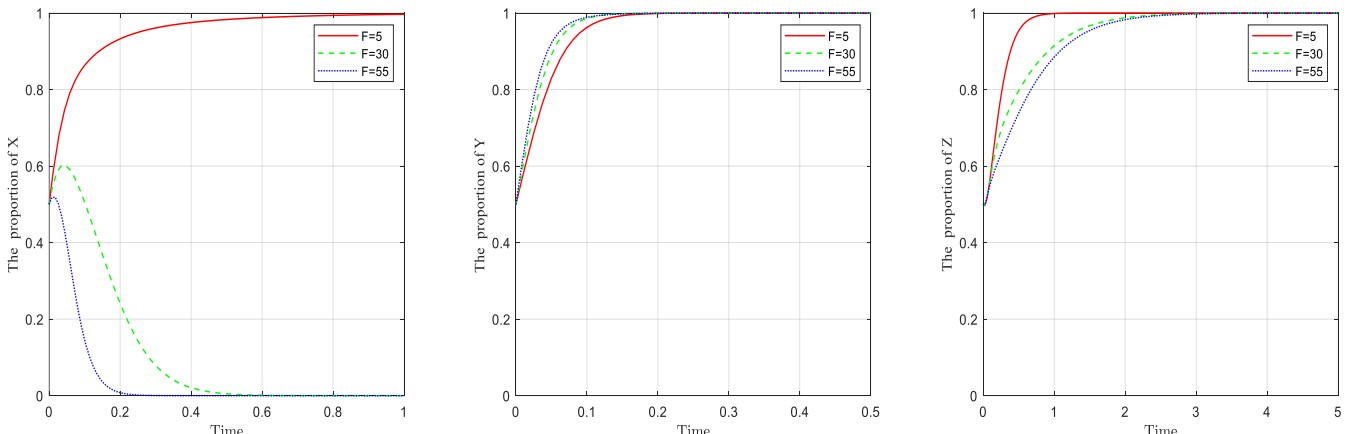

**Figure 9.** The effect of $F$ on the evolution of the system to an asymptotic steady state.

5.2.5. Sensitivity Analysis of the Cost of Active Consumer Regulation for System Evolution

Let the values of the remaining influencing factors be fixed and the active supervision cost of consumers be taken as $C_{C1} = 4, 8, 12$. The system evolution result is shown in Figure 10 below. As shown in the figure, with the increase in $C_{C1}$, the rate of convergence of e-commerce platforms to non-algorithmic price discrimination gradually decreases, and the rate of convergence between local governments and consumers to active supervision also gradually decreases. When $C_{C1}$ increases to a certain threshold, it will lead consumers to choose negative regulatory strategies. This shows that the cost of active supervision of consumers only affects the convergence speed of the platform and the local government. As the remaining values of the influencing factors remain unchanged, it is not possible to change the strategy choices of either party. For consumers themselves, when the platform chooses non-algorithm price discrimination, if their regulatory costs exceed the incentives given by the local government $S$, they will not be able to cooperate with the government to actively supervise. At this time, the local government can reduce the cost of active supervision of consumers by sharing price data information, encouraging the establishment of third-party price comparison platforms, etc. This will give consumers the motivation to actively coordinate supervision.

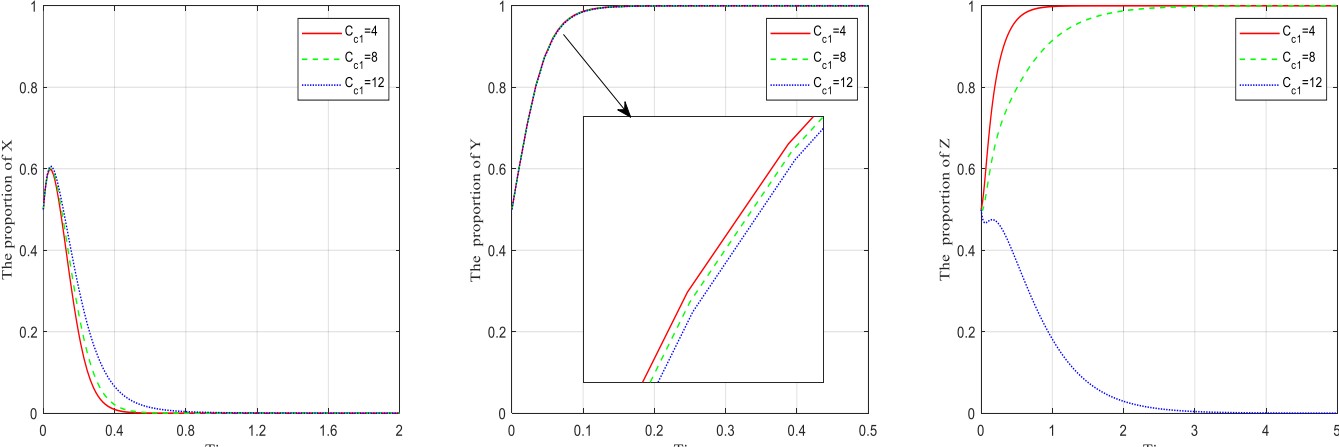

**Figure 10.** The effect of $C_{C1}$ on the evolution of the system to an asymptotic steady state.

5.2.6. Sensitivity Analysis of Consumers' Active Collaborative Regulation Reward for System Evolution

Let the parameter values of the remaining influencing factors be fixed, and consider the bonus to be $S = 5, 10, 45$. The system evolution result is shown in Figure 11 below. It can be seen from the figure that, when $S$ is at a low level, the rate of convergence of e-commerce platforms to non-algorithmic price discrimination strategies is low. Local governments have a stronger willingness to actively supervise, and consumers choose negative regulatory strategies due to the lack of regulatory motivation. With the gradual increase in $S$, the convergence rate of e-commerce platforms accelerates. At this time, the willingness of local governments to regulate is slightly weakened, but they still choose to actively supervise. Appropriate bonuses will prompt consumers to turn to active regulatory strategies. When $S$ increases to a certain threshold, although the high bonus brings strong regulatory momentum to consumers, it also increases the regulatory cost of local governments. Their strategy choices will fall into cyclical shocks, and wavering decisions will also make the strategic choices of e-commerce platforms change cyclically. Thus, the incentive given by local governments to consumers for active collaborative supervision does not linearly increase with the increase in the reward. The bonus threshold should be reasonably set according to its own costs and consumers' costs, and, at the same time, the local governments should actively improve the relevant reward rules. This will enhance consumers' willingness to actively cooperate with supervision.

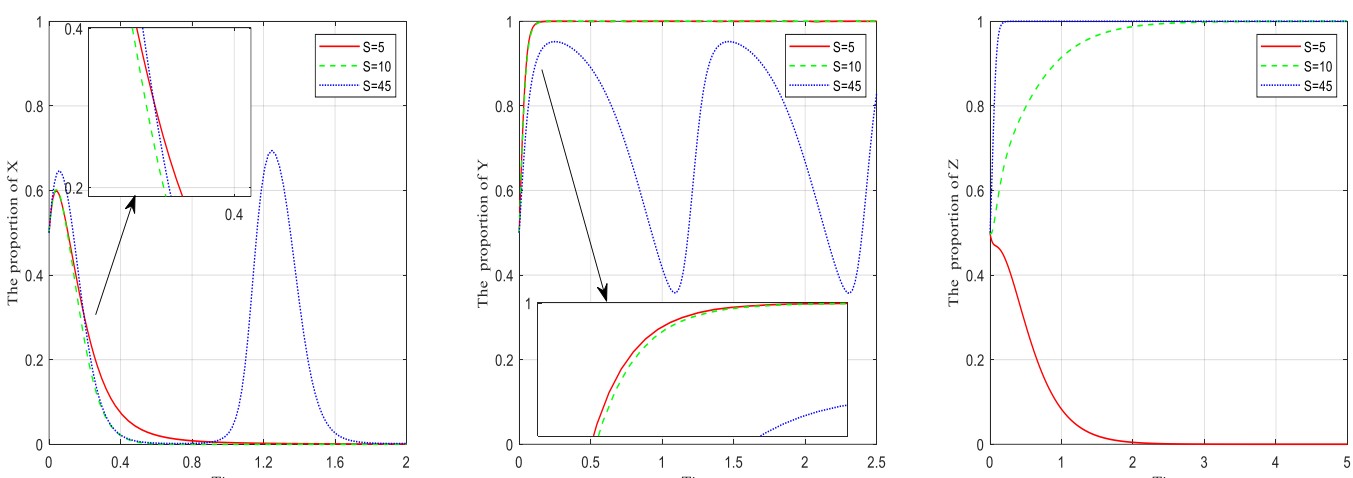

**Figure 11.** The effect of $S$ on the evolution of the system to an asymptotic steady state.

## 6. Conclusions and Recommendations

### 6.1. Conclusions

This paper constructs a three-party evolutionary game model for e-commerce platforms, local governments and consumers. To a certain extent, it broadens the perspective of existing research in the field of algorithmic price discrimination. We conducted analyses on the asymptotic stability of the system under different scenarios, the corresponding strategy choices of the three parties and comparative experiments by controlling the values of the variables. In doing so, we explored the influence of various factors on the evolution of the system. Based on the above model derivation and simulation experimental results, the following conclusions are drawn:

(1) When the sum of the fines levied and the consumer network externalities lost by the platform is greater than the excess grey income, which is obtained by the e-commerce platform from algorithmic price discrimination, it can promote the platform to set reasonable prices. The initial willingness of the three-party strategy choice will not change the final evolutionary result, but a higher initial willingness will promote the system to evolve to a stable state faster.

(2) When the degree of algorithmic price discrimination on e-commerce platforms increases, although the rate of active collaborative supervision between local governments and consumers accelerates, the rate of platforms tending towards non-algorithmic price discrimination slows down. Therefore, it is necessary to cooperate with the media to increase the length of reporting on platform algorithm price discrimination, to improve consumers' price fairness preferences [44] and to publicize the government's determination to rectify the chaos of algorithm price discrimination. To curb the short-sighted behaviour of platform algorithm price discrimination, it is also necessary to strengthen industry self-discipline and safeguard public rights and interests.

(3) The increase in excess grey gains from algorithmic price discrimination on e-commerce platforms will significantly change the strategic choices of the platforms. When the potential excess grey income exceeds a certain threshold, the platforms will engage in algorithmic price discrimination for short-term gain, thereby ignoring the long-term brand reputation benefits brought by consumer network externalities. Therefore, government regulators need to set up clear rules for punishments to quantify the losses caused by the short-sighted behaviour of the platforms and to make the platforms aware of their social responsibilities [45]. In addition, consumers should actively use the advantages of self-media and network externalities to maintain their right to pursue price fairness.

(4) The greater the probability that actively regulating local governments find algorithmic price discrimination, the faster the rate at which e-commerce platforms choose non-algorithmic price discrimination. However, local governments alone are limited in their ability to regulate and still need the active collaboration of consumers. Therefore, government regulatory authorities need to unblock complaint channels and establish a long-term mechanism for consumer feedback [46]. Positive consumer synergy is well received, which, in turn, increases the willingness of consumers to actively regulate.

(5) The strength of local government fines against platforms for algorithmic price discrimination is positively correlated not only with the rate of active regulation but also with the rate at which platforms converge to non-algorithmic price discrimination. The rate at which consumers opt for active regulation is negatively correlated with the strength of fines due to their free-rider mentality [47]. Therefore, government regulatory authorities should also choose appropriate rewards and punishments, improve reward and punishment mechanisms, curb platform algorithm discrimination and give consumers the motivation to actively coordinate supervision.

(6) Consumers' strategy choices are extremely sensitive to the impact of their active regulatory costs. The lower the regulatory cost, the higher the return during active supervision, and the more it will prompt them to choose an active collaborative

strategy. At the same time, it can effectively increase the rate of platform choice of non-algorithmic price discrimination. Therefore, on the one hand, the government regulatory department shares price data information, reduces the degree of market information asymmetry and promotes price transparency [48], while, on the other hand, consumers should also continuously improve their own price comparison capabilities and make reasonable use of third-party price comparison platforms to reduce regulatory costs.

(7)　The amount of money awarded for active co-regulation by consumers is more disruptive to the evolution of the system. A low incentive for consumer regulation when the bonus is too low and an imbalance in local government revenues and expenditures when the bonus is too high will lead to cyclical shocks and a cyclical change in the e-commerce platform. Therefore, the amount of money received by consumers should not be too high or too low but should be within a certain threshold, taking into account the benefits to the government and the costs to consumers. This will not only effectively regulate the platform algorithm price-discriminatory behaviour but also achieve active co-regulation between the local government and consumers and, thus, a win–win situation for all parties.

### 6.2. Recommendations

Curbing the short-sighted behaviour of algorithmic price discrimination on e-commerce platforms is important for maintaining consumer fairness, enhancing consumer trust and, thus, promoting the sustainable development of e-commerce platforms. Based on the above research, the following recommendations are made from the perspectives of e-commerce platforms, local governments and consumers, respectively:

(1)　E-commerce platforms should focus on brand reputation, enhance social responsibility and strengthen industry self-regulation. In a highly competitive market environment, e-commerce platforms must focus on consumer word-of-mouth and establish a good brand reputation if they are to have longevity. In addition, the e-commerce platform should consciously fulfil its social responsibility and actively cooperate with local government supervision. At the same time, industry associations should improve the industry norms for e-commerce platforms and strengthen industry self-discipline. This will create a good atmosphere for the development of the e-commerce industry.

(2)　Local governments should improve reward and punishment mechanisms, increase publicity efforts and open up channels for consumer complaints; establish a sound complaint mechanism and accountability system for algorithmic price discrimination on e-commerce platforms and set the reasonable rewards and penalties within a threshold; place publicity announcements on government websites, public websites and third-party media to alert e-commerce platforms of punishments for algorithmic price discrimination; promote information technology and visualization as a means of big data regulation; and reduce the level of information asymmetry in the e-commerce market. Meanwhile, the complaint channels will reduce the cost of co-regulation by consumers.

(3)　Consumers should enhance their awareness of their rights and actively improve their ability to compare prices. They should establish a correct consumer mindset, enhance their awareness of their rights and be brave enough to use legal means to defend their rights and interests in the face of unfairness. In addition, they should actively make use of third-party price comparison platforms and others to enhance price comparison capabilities. They should be adept at harnessing the power of public opinion, moving from passive acceptance to active participation and co-regulation.

### 6.3. Limitations and Further Research

There are still limitations in this study, which only concern the evolutionary game between e-commerce platforms, local governments and consumers. The influence of merchants and third-party media on stakeholders' participation has not been consid-

ered. Therefore, in future research, more stakeholders can be considered as participants, and a multi-stakeholder evolutionary game model can be constructed for a more comprehensive analysis.

**Author Contributions:** J.L. is mainly responsible for designing the outline and writing guidance; X.X. is mainly responsible for writing the article, designing the models and conducting the simulation experiments; Y.Y. is responsible for article polishing. All authors have read and agreed to the published version of the manuscript.

**Funding:** This research was supported by the 2022 Heilongjiang Philosophy and Social Science Research Planning Project, grant number 22SHE416; the 2022 Harbin University of Commerce "Innovation" Project Support Program, grant numbers XW0093 and XW0145; and the Harbin Science and Technology Plan Self-Financing Project, grant number ZC2022ZJ014004.

**Institutional Review Board Statement:** Not applicable.

**Informed Consent Statement:** Not applicable.

**Data Availability Statement:** Not applicable.

**Conflicts of Interest:** The authors declare no conflict of interest.

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
