# Peer review of "Research on the Regulation of Algorithmic Price Discrimination Behaviour of E-Commerce Platform Based on Tripartite Evolutionary Game"

_sustainability, doi:10.3390/su15108294_

Round 1

Reviewer 1 Report

Following changes have been made before consideration for publication

1. In the Introduction Section, motivation, objective, and contribution need to be presented clearly.

2. What the different parameters are taken into consideration, needs to be specified clearly.

3. In the methodology section, nothing is explained clearly. You have to specify, which and how you have done this work with proper examples, models, and explanations. 

4. Performance measures should be a consideration in another section.

5. Please specify the performance measures and its significance in this problem.

6. In the conclusion section, drawbacks and future scope, need to be highlighted. 

Author Response

Thank you very much for your comments and professional advice!

Reviewer 2 Report

Dear Authors,

Please take into consideration the following issues:

1. The TITLE seems appropriate.

2. The type of research method should be specified in the ABSTRACT alongside the aim(s) of the paper.

3. The aim(s) of the research should be clearly presented in the INTRODUCTION. Also, it should be structured in a manner suitable to better emphasize the context of the paper and the relevance of the subject.

4. In general, the structure of a scientific paper comprises the following main sections: INTRODUCTION, LITERATURE REVIEW, RESEARCH METHODOLOGY, RESULTS, DISCUSSION and CONCLUSIONS. Thus, the paper may be restructured.

5. Statements such as "At present, many scholars have studied various aspects of algorithmic price discrimination on e-commerce platforms, including the mechanisms at play and behavioural regulation measures" should be related to specific references.

6. In CONCLUSIONS, the authors should better outline why the outcomes of their study are relevant from a scientific point of view. Also, they should emphasize the relationships with other studies.

7. The use of the English language should be improved.

Good luck!

Author Response

(The authors gave the same response as above.)

Reviewer 3 Report

Reviewer

Comments

The abstract provides a clear overview of the potential negative impacts of algorithms and big data technologies in e-commerce platforms and proposes measures to promote sustainable development and regulate algorithmic price discrimination.

However, it could benefit from more specific examples and elaboration on the proposed measures.

Introduction

It would be useful to provide more information on the methodology and approach taken in the study and the contributions and novelty of the research.

The literature review provides a comprehensive overview of the research on algorithmic price discrimination on e-commerce platforms. Overall, the literature review provides a good overview of the current research on algorithmic price discrimination on e-commerce platforms, including the mechanisms involved and behavioural regulation measures.

By addressing the following points, the literature review could be improved to better communicate its message to readers:

Structure: The review is structured as a single paragraph, making it difficult for readers to follow the argument. It would be helpful to break it down into smaller sections or paragraphs.

Citations: Some of the statements in the review are not cited, making it difficult for readers to verify the claims being made. It is important to provide proper references to support each point.

Clarity: Some of the sentences are long and complex, making it difficult to understand the key points being made. It would be helpful to simplify the language and use shorter sentences to make the review more accessible to readers.

Focus: The review covers a lot of ground, but it would benefit from a clearer focus. The review could be more effective if it were to hone in on a specific aspect of the research, such as the impact of algorithmic price discrimination on consumer behaviour or the most effective regulatory measures.

Original Contribution: While the review provides a good summary of existing research, it would benefit from a clearer statement of its original contribution to the field. It would be helpful to explain how the review adds to the existing knowledge and what gaps in the research it aims to address. 

Language

However, the language can be simplified to make it more reader-friendly. For example, instead of using the term "oppressed party," the text can use "consumers who have little power to defend their rights." Moreover, the text could be improved by breaking down some of the longer sentences into shorter, simpler ones. This will make the text more readable and accessible to a wider audience. 

Conclusion:"The findings and conclusion of the study are insightful and provide useful guidance for e-commerce platforms, local governments, and consumers to work collaboratively in regulating algorithmic price discrimination and safeguarding public rights and interests.

However, the findings and conclusions are not supported by similar research.

It will be better to support the findings and conclusions of the paper with findings of similar research published.

Author Response

(The authors gave the same response as above.)

Reviewer 4 Report

Very long sentences. Some sentences are four lines or more. This makes the paper hard to read and follow.

Very long paragraphs. To improve the flow and readability of the paper, each page should have 3-5 paragraphs

I do not know why you call the 4 assumptions hypotheses. Are you testing them? Why are they written in italic?

For the replication dynamics equation, why do you use the same function, F? Do the three parties have the same function?

The reader has to wait till section 5 to understand this is a numerical simulation. The abstract list several factors affecting the algorithm price discrimination, implying the study was done with market data.

Author Response

(The authors gave the same response as above.)

Round 2

Reviewer 4 Report

I still do not think you should call your assumptions hypotheses.